# Foveal Avascular Zone Enlargement as a Risk Factor for Early Retinal Ganglion Cell Dysfunction in Glaucoma Suspects

**DOI:** 10.3390/diagnostics15162103

**Published:** 2025-08-21

**Authors:** Andrew Tirsi, Kashviya Suri, Samuel Potash, Joby Tsai, Danielle Kacaj, Vasiliki Gliagias, Nicholas Leung, Timothy Foster, Rushil Kumbhani, Derek Orshan, Daniel Zhu, Celso Tello

**Affiliations:** 1Department of Ophthalmology, Manhattan Eye, Ear, and Throat Hospital, Northwell Health, New York, NY 10065, USA; ctello@northwell.edu; 2Donald and Barbara Zucker School of Medicine at Hofstra/Northwell, Hempstead, NY 11549, USA; ksuri@northwell.edu (K.S.); dak264@cornell.edu (D.K.); vgliagias@northwell.edu (V.G.); nleung1@northwell.edu (N.L.); tfoster5@northwell.edu (T.F.); rkumbhani1@pride.hofstra.edu (R.K.); dzhu@northwell.edu (D.Z.); 3Albert Einstein College of Medicine, Bronx, NY 10461, USA; samuel.potash@einsteinmed.edu; 4Department of Ophthalmology, Broward Health, Deerfield Beach, FL 33064, USA; jtsai@browardhealth.org; 5Department of Ophthalmology, Larkin Community Hospital, Miami, FL 33431, USA; derekorshan@gmail.com

**Keywords:** glaucoma, glaucoma suspect (GS), pattern electroretinogram (PERG), optical coherence tomography angiography (OCTA), foveal avascular zone (FAZ), retinal ganglion cell (RGC)

## Abstract

**Background/Objectives:** The aim of this study was to evaluate the relationship between foveal avascular zone (FAZ) enlargement, retinal ganglion cell (RGC) dysfunction, and structural retinal measurements in glaucoma suspects (GS), using pattern electroretinogram (PERG) and optical coherence tomography angiography (OCTA) parameters. **Methods:** Thirty-one eyes (20 subjects) of GS status underwent comprehensive ophthalmologic evaluation including steady-state PERG, optical coherence tomography (OCT), and OCTA. FAZ area was measured using ImageJ software (version 1.54p), and PERG parameters (Magnitude, MagnitudeD, and MagnitudeD/Magnitude ratio) were analyzed. Partial correlation analyses were performed to assess associations between FAZ area, PERG parameters, and structural metrics including retinal nerve fiber layer (RNFL), ganglion cell layer–inner plexiform layer (GCL + IPL), and macular thickness. **Results:** After controlling for age, sex, central corneal thickness (CCT), intraocular pressure (IOP), and spherical equivalent, partial correlation analysis showed that FAZ area was significantly associated with both lower Magnitude (*r* < −0.503, *p* < 0.05) and MagnitudeD (*r* < −0.507, *p* < 0.05) values. PERG parameters were significantly correlated with superior and average RNFL thickness, as well as superior and superior temporal GCL + IPL thickness. FAZ area was significantly associated with multiple GCL + IPL and macular thickness sectors, but not with RNFL thickness. **Conclusions:** FAZ enlargement is significantly associated with RGC dysfunction and inner retinal layer thinning in GS.

## 1. Introduction

Glaucoma is the leading global cause of irreversible blindness [1]. Primary Open-Angle Glaucoma (POAG) is a chronic optic neuropathy characterized by the progressive degeneration of retinal ganglion cells (RGCs) [2]. By the time these patients first develop visual field loss, 35–45% of RGCs have already been lost [3]. This RGC loss is also seen in glaucoma suspects (GS), preceding both visual field loss and OCT changes [3]. To date, lowering Intraocular Pressure (IOP) is the only modifiable risk factor that has been effective in reducing glaucomatous progression [4,5,6,7]. While IOP-lowering therapies (pharmacologic, laser, or surgical) remain the gold standard for POAG management, disease progression in patients with well-controlled IOP suggests that additional, IOP-independent mechanisms may contribute to glaucomatous damage [8]. Normal-tension glaucoma (NTG) has long been associated with vascular etiologies [9]. Various studies have indicated systemic hypotension, particularly nocturnal hypotension, as an important risk factor for glaucomatous optic neuropathy (GON) and visual field defect progression in NTG [10,11]. It is thought that systemic hypotension and vascular dysregulation lead to GON via reduced ocular perfusion pressure (OPP) and ocular blood flow (OBG) instability, respectively [12,13,14,15,16,17,18,19,20,21].

Emerging evidence suggests that vascular factors may also play a role in POAG. Multiple studies and meta-analyses have demonstrated that patients with POAG exhibit significantly lower OPP and greater 24 h OPP fluctuations, both of which are associated with disease progression [22,23,24]. Optical coherence tomography angiography (OCT-A) has revealed reduced peripapillary and optic nerve head vessel density in POAG, correlating with retinal nerve fiber layer thinning and aiding in early disease detection [25,26,27]. Additionally, Doppler imaging studies have consistently shown decreased peak systolic and end-diastolic velocities, along with increased vascular resistance, in the ophthalmic artery and central retinal artery of POAG patients, further implicating compromised ocular blood flow in glaucomatous damage [28,29]. One area of growing interest in this vascular context is the foveal avascular zone (FAZ), a capillary-free region at the center of the macula, whose area can be measured using OCT-A. Studies have shown that FAZ area is increased in patients with POAG [30]. Additionally, larger FAZ areas have been found to be associated with poorer visual acuity in moderate–advanced POAG patients [31]. Importantly, FAZ area has demonstrated diagnostic utility comparable to retinal nerve fiber layer (RNFL) and ganglion cell layer–inner plexiform layer (GCL + IPL) thickness, both of which undergo thinning in glaucoma [32]. Moreover, FAZ enlargement may be reversible following IOP-lowering interventions, suggesting that it could serve as a modifiable vascular marker of disease [33].

Pattern electroretinogram (PERG) is a non-invasive electrophysiological test that can assess RGC function using a reversing checkerboard stimulus [34]. It offers high repeatability and reproducibility [35,36] and has demonstrated strong diagnostic potential in glaucoma [37,38]. It is able to detect RGC dysfunction years before structural changes in average retinal nerve fiber layer (ARNFLT) thickness are seen on OCT [39] or before visual field loss is evident on standard automated perimetry [37]. Cross-sectional studies have also consistently reported PERG deficits in GS [40,41,42]. Our lab has previously discovered positive correlation between PERG parameters and average retinal vessel diameter in pre-perimetric glaucoma patients [43], suggesting that PERG abnormalities have a vascular etiology [44]. No studies have yet compared the relationship between PERG and FAZ area in GS.

Our lab has previously shown that in GS, PERG parameters correlated with structural changes such as RNFL and GCL + IPL thinning, as well as optic nerve head morphology [45]. Given that PERG can detect RGC dysfunction years before field loss occurs, it offers a unique opportunity to explore early vascular compromise [46].

This study aimed to investigate whether GS with RGC dysfunction, exhibited FAZ area enlargement and whether this enlargement correlated with PERG abnormalities and structural retinal thinning. If confirmed, FAZ area could serve as an additional, non-invasive marker of macular microvascular integrity in early glaucoma and GS, potentially expanding our methods for identifying and managing at-risk individuals before irreversible damage occurs.

## 2. Materials and Methods

### 2.1. Study Design

All eligible subjects were recruited from Manhattan Eye, Ear and Throat Hospital (MEETH) ophthalmology practice and twenty consecutive subjects (31 eyes) were included in this cross-sectional study. All participants underwent a complete ophthalmologic examination, including slit lamp biomicroscopy, Goldmann tonometry, standard automated perimetry (Humphrey Field Analyzer [HFA] II 24-2 Swedish Interactive Threshold Algorithm [SITA]-Standard strategy (Carl Zeiss Meditec Inc., Dublin, CA, USA)) [47], Spectral Domain (SD)-OCT Zeiss Cirrus High Definition (HD)-OCT (Carl Zeiss Meditec, Dublin, CA, USA), and PERG (Diopsys Inc., Cedar Knolls, NJ, USA). This study was approved by the Institutional Review Board of Northwell Health System (IRB #18-0397). Written informed consent was obtained from all subjects, and the study followed the tenets of the Declaration of Helsinki.

### 2.2. Inclusion Criteria

Participants were recruited according to the following criteria: the presence of a suspicious glaucomatous optic nerve head appearance (increased cup to disk ratio > 0.4, neuroretinal rim thinning, notching or excavation) and a normal HFA 24-2 SITA-standard test at the baseline visit. Participants 18–80 years of age, with best corrected visual acuity (BCVA) better or equal to 20/40 (Snellen), any IOP level, and any type of angle were enrolled in this study. Other inclusion criteria were spherical refraction within ±6.0 D, cylinder correction within 3.0 D, documented and repeatable normal HFA 24-2 using the HFA 24-2 Swedish Interactive Threshold Algorithm (SITA) and defined by Glaucoma Hemifield Test (GHT) “within normal limits” or pattern standard deviation within 95% confidence limits, mean deviation (MD) normal values ≥ −2 dB, and individuals with no IOP lowering treatment at the time of enrollment.

### 2.3. Exclusion Criteria

Participants with prior anterior or posterior segment intraocular surgery, ocular trauma, ocular or systemic conditions that may affect the optic nerve head, retinal structure, or function (e.g., ischemic optic neuropathy, optic neuritis, papilledema, and corneal and retinal diseases) were not included in this study. Other exclusion criteria involved individuals with prior intraocular surgery in the study eye, except for uncomplicated cataract extraction with posterior chamber intraocular lens (IOL) implant and no escape of vitreous to the anterior chamber that was performed less than a year before enrollment, and individuals with unreliable HFA visual field results, with fixation losses, false positive rate, and false negative rate, of more than 20% each. Based on the inclusion/exclusion criteria, nine eyes were removed from the study. One eye was diagnosed with epiretinal membrane and eight were diagnosed with early glaucoma.

### 2.4. Tests Performed

#### 2.4.1. Tonometry

Goldmann applanation tonometry was used to measure IOP and two consecutive measurements were obtained differing by 1 mm Hg or less. The average of the 2 readings was recorded as the IOP measurement.

#### 2.4.2. Central Corneal Thickness Measurement

Many reports have shown that thinner-than-average corneas may underestimate the true IOP whereas thicker-than-average corneas may overestimate it. This effect has been found on the order of 1 mmHg correction for every 25 µm deviation from a central corneal thickness (CCT) of 550 µm. The handheld ultrasound PachPen (Accutome, Inc., Malvern, PA, USA) was used to determine CCT, and an average of 3 measurements was recorded. CCT measurements were considered in IOP estimations and RNFL thickness measurements.

#### 2.4.3. Visual Field Testing

All participants in this study had prior experience with VF examination, using a Humphrey perimeter (HFA II, Carl Zeiss Meditec, Inc., Dublin, CA, USA). This is the standard test used in ophthalmology for VF assessment and has been described elsewhere. In short, the patient is required to fixate on a central point and push a button whenever they see a flashing light. The SITA 24-2 and SITA 10-2 standard strategies and Global Indices will be used including MD, Pattern Standard Deviation, and Visual Field Indices (VFI), all provided on a statistical summary. Using HFA SITA 24-2 results, only participants with VFs corresponding to stage 0 (no VF losses) following the Glaucoma Staging System (GSS 2) will be considered.

#### 2.4.4. OCT Angiography (OCTA)

OCTA measurements of the macula for each eye were obtained using a Zeiss Cirrus HD-OCT 5000 with AngioPlex device (Carl Zeiss Meditec, Inc., Dublin, CA, USA). OCTA provides angiographs from the superficial, intermediate, and deep vascular complex. Scans measuring 3 × 3 mm (superficial retinal complex) were used in the study. The superficial vascular complex was used to obtain the macular FAZ area. The FAZ area was measured manually using ImageJ software. The intraclass correlation coefficient was 0.99 indicating a high degree of reliability between 2 raters (AT, JT) (Figure 1).

#### 2.4.5. Optical Coherence Tomography (OCT) Parameters

All OCT parameters were measured using the Zeiss Cirrus HD-OCT 5000. The Optic Disc Cube 200 × 200 scan was used to obtain the RNFLT around the optic nerve head in four quadrants: Superior (S), Inferior (I), Nasal (N) and Temporal (T). Comparisons were made among the different quadrants of the RNFL. Average RNFLT was also obtained and compared with a normative database.

The macular thickness map protocol with a macular cube 512 × 128 combo was used to obtain macular thickness. The macula was divided into eleven sub-sectors and measurements of the thickness in each sector was calculated. The macular thickness sectors are Superior Outer (SO), Superior Inner (SI), Nasal Outer (NO), Nasal Inner (NI), Inferior Outer (IO), Inferior Inner (II), Temporal Outer (TO), Temporal Inner (TI), Thickness Central Subfield, Volume Cube, and Thickness Average Cube.

The SD-OCT Macular Cube 200 × 200 test was used to calculate the average GCL + IPL thickness, minimum GCL + IPL thickness, and regional GCL + IPL thicknesses in the six sectors, being Superior (S), Superior Nasal (SN), Superior Temporal (ST), Inferior (I), Inferior Nasal (IN) and Inferior Temporal (IT). Comparisons among the six sectors were made to determine which sector shows the most localized glaucomatous changes. The average GCL + IPL thickness is a number averaged from the six sectors. The minimum GCL + IPL shows the lowest GCL + IPL thickness.

#### 2.4.6. Pattern Electroretinogram (PERG) Measurements

RGC dysfunction was measured using the diagnostic Diopsys NOVA-PERG device and the procedure has been described previously [45,48,49]. In short, a subject was seated in front of the monitor and three sensors were placed on the lower lids and forehead and connected to the device. A visual stimulus was shown to the subject on the monitor while the device measured the electrical signals generated by his/her eyes.

For each eye, three PERG measures (Magnitude [Mag], MagnitudeD [MagD], and MagnitudeD/Magnitude [MagD/Mag] ratio) were collected. Mag (in µV) represents the amplitude or the signal strength at the specific reversal rate of 15 Hz, in the frequency domain, while MagD indicates the amplitude of the PERG signal and its relation to phase variability throughout the waveform recording. A recording that is in-phase produces a MagD value close to that of Magnitude and an out-of-phase recording produces a MagD value significantly lower than that of Mag. MagD/Mag ratio is a ratio that is a within-subject representation of the phase consistency of PERG. The PERG test was repeated, if categorized as non-reliable, for a maximum of 3 times. After 3 unsuccessful attempts, the patient was excluded from the study (Figure 2).

### 2.5. Statistical Analysis

All statistical analysis was performed using the IBM Statistical Product and Service Solutions (SPSS) Statistics software Version 26. For all variables of interest, outliers with values ≥ 3 standard deviations from the mean were excluded from the analyses. Descriptive statistics were used to evaluate continuous and demographic data. Mean and standard deviation values were determined for each PERG parameter (Mag, MagD, MagD/Mag ratio), HFA SITA standard (24-2) global indices, FAZ area, all RNFLT variables, macular thickness variables, and GCL + IPL thickness variables. Partial correlation analysis was used to test the correlation of the PERG parameters and FAZ area with the RNFL, GCL + IPL, and macula. This study tested whether FAZ area was enlarged and if it was associated with PERG parameters, RNFL, GCL + IPL, and macular thickness. An alpha level of *p* < 0.05 was used in this study.

### 2.6. Mediation Analysis

A mediation analysis was conducted on PERG parameters using the PROCESS macro (version 3.5) for SPSS, developed by Andrew Hayes [50]. We performed a mediation analysis with MagD as X, FAZ area as M, and Superior GCL + IPL Thickness as Y. This analysis was designed to explore whether the effect of an independent variable (X) on a dependent variable (Y) occurs directly, or indirectly through a third variable, known as the mediator (M) (Figure 3). In this framework, the direct effect refers to the influence of X on Y without accounting for the mediator. The indirect effect, on the other hand, captures the pathway in which X affects M, which subsequently influences Y. This indirect effect is calculated by multiplying the strength of the relationship between X and M with the strength of the relationship between M and Y. To determine the statistical significance of both the direct and indirect effects, the analysis employed 5000 bootstrapped samples and reported 95% confidence intervals. The strengths and limitations of this mediation approach have been previously discussed in the literature by Yau et al. (2017), who applied a similar model to examine how insulin sensitivity and inflammation mediate the relationship between fitness and cerebrovascular health in adolescents [51]. Our team has applied mediation analysis in prior work to interpret vascular and functional relationships in glaucoma [52].

## 3. Results

A total of 31 eyes (20 subjects; 14 Females, 6 Males) with a mean age of 59.89 ± 14.20 years were included in this study. All the participants were enrolled as part of a longitudinal IRB approved study and met the inclusion criteria. Demographic data are reported in Table 1. Mean central corneal thickness (CCT) was 551.67 ± 29.22. and mean intraocular pressure (IOP) was 18.29 ± 4.46 mmHg, both of which fell within normal limits. The mean spherical equivalent was −0.531 ± 2.34.

The mean baseline values obtained on all testing modalities used in this study are reported in Table 1 and Table 2. The baseline mean HFA MD 24-2 was 0.21 ± 0.94 dB. On steady state pattern electroretinogram (ssPERG), the mean Mag was 1.56 ± 0.43, mean MagD was 1.34 ± 0.58, and the mean MagD/Mag ratio was 0.83 ± 0.11. When analyzing the OCT-A images, the mean FAZ area was 0.32 ± 0.11 mm^2^ and the mean RNFL thickness for the Superior, Temporal, Inferior, Nasal, and overall Average zones are reported in Table 1. GCL + IPL and macular thickness measurements for each sector, as well as the Average and Minimum, are reported in Table 2.

Results of a partial correlation analysis between FAZ area and PERG amplitude are reported in Table 3. This analysis controlled for Age, Sex, CCT, IOP, and Spherical Equivalent. It was observed that the Mag and MagD measurements displayed a significant correlation with FAZ area (*r* < −0.503, *p* < 0.028 and *r* < −0.507, respectively). The ratio of Mag/MagD exhibited no significant correlation with the FAZ area measurements.

Partial correlation analysis of RNFL thickness, FAZ area, and PERG parameters were reported in Table 4. Mag, MagD, and MagD/Mag ratio were all found to be significantly correlated with the Superior Sector of RNFL (*r* > 0.759, *r* > 0.807, and *r* > 0.628, respectively) and the Average RNFL (*r* > 0.537, *r* > 0.600, and *r* > 0.530, respectively), but not with the Temporal, Inferior, or Nasal sectors. The FAZ area exhibited no significant correlations with RNFL measurements.

Partial correlation analysis of the GCL + IPL, FAZ Area, and PERG Parameters are reported in Table 5. Mag was observed to be significantly correlated with the Superior (*r* > 0.586), Superior Temporal (*r* > 0.556), and Inferior Nasal GCL + IPL thickness sectors (*r* > 0.496). No significant correlation was seen with Mag and the Superior Nasal, Inferior, or Inferior Temporal Sectors of the GCL + IPL. In addition, no significant relationship was observed between the Mag value and the Average GCL + IPL thickness or the Minimum GCL + IPL thickness.

MagD was significantly correlated with the Superior (*r* > 0.618) and Superior Temporal (*r* > 0.621) GCL + IPL thickness sectors, but not significantly correlated with the Superior Nasal, Inferior, or Inferior Temporal sectors. However, the MagD value was significantly correlated with the Average GCL + IPL and the Minimum GCL + IPL values (*r* > 0.490 and *r* > 0.540, respectively).

The MagD/Mag ratio was significantly correlated with the Superior (*r* > 0.527) and Superior Temporal (*r* > 0.599) GCL + IPL thickness sectors, but not the Superior Nasal, Inferior, Inferior Nasal, or Inferior Temporal Sectors. The MagD/Mag ratio was also significantly correlated with the Average GCL + IPL thickness and the Minimum GCL + IPL thickness values (*r* > 0.595 and *r* > 0.704, respectively). The Superior and Superior Temporal GCL + IPL thickness sectors were consistently significantly correlated with PERG parameters of Mag, MagD, and MagD/Mag ratio.

FAZ area was significantly correlated with the Superior, Superior Nasal, Superior Temporal, Inferior, and Inferior Nasal sectors of GCL + IPL, but not the Inferior Temporal sector. In addition, the FAZ area exhibited no significant correlation with the Average GCL + IPL or the Minimum GCL + IPL thickness values.

Partial correlation analysis of macular thickness, FAZ area, and PERG parameters was also performed (Table 6). A significant relationship between Mag and the Macular Thickness of the Superior Inner, Nasal Outer, Nasal Inner, Inferior Inner, and Temporal Inner sectors of the macula was observed. No significant relationship was observed between Mag and the Superior Outer, Inferior Outer, or Temporal Outer macula sectors. Likewise, no significant correlation was observed between the Thickness of the Central Subfield, Volume Cube value, or Thickness Average Cube measurements.

The MagD measurement was significantly correlated with the Superior Inner (*r* > 0.512), Inferior Inner (*r* > 0.481), and Temporal Inner (*r* > 0.625) macula sectors. No significant correlation was observed between MagD and the Superior Outer, Nasal Outer, Nasal Inner, Inferior Outer, or Temporal Outer macula sectors, nor was any significant correlation determined with the Thickness Central Subfield, Volume Cube, or Thickness Average Cube measurements. The MagD/Mag ratio showed a significant correlation only with the Temporal Inner macula sector (*r* > 0.509).

The FAZ area was found to be significantly correlated with the Superior Outer (*r* < −0.555), Superior Inner (*r* < −0.580), Nasal Outer (*r* < −0.708), Nasal Inner (*r* < −0.655), Inferior Inner (*r* < −0.562), Temporal Inner (*r* < −0.512) macula sectors in addition to the Thickness Central Subfield (*r* < −0.761), Volume Cube (*r* < −0.631), and Thickness Average Cube (*r* < −0.471). No significant correlation was observed between FAZ and the Inferior Outer or Temporal Outer macula sectors.

Overall, PERG measurements consistently correlated significantly with the Inner sector values while FAZ Area exhibited a significant correlation with every metric except for the Inferior and Temporal Outer sectors.

Mediation analysis was conducted to assess whether FAZ Area mediates the relationship between MagD and Superior GCL + IPL Thickness (Figure 4). The analysis revealed a significant direct effect of MagD on Superior GCL + IPL Thickness (*b* = 4.93, *p* < 0.05). Additionally, a significant indirect effect was observed through FAZ Area (*b* = 2.71, *p* < 0.05). Specifically, MagD was negatively associated with FAZ Area (*b* = −0.15, *p* < 0.05), and FAZ Area was negatively associated with Superior GCL + IPL Thickness (*b* = −18.98, *p* < 0.05).

## 4. Discussion

In this study, we report a significant relationship between PERG parameters and FAZ area, suggesting a vascular etiology to the RGC dysfunction in GS. Specifically, we observed that individuals with larger FAZ areas exhibited greater RGC dysfunction, as measured by PERG. Significant negative correlations were found between FAZ area and both Mag and MagD (Table 3), suggesting that FAZ enlargement may reflect early functional compromise of the RGC complex, even in the absence of overt structural loss in GS.

These results support that FAZ area has significant implications for PERG parameters. Enlargement of the FAZ is often interpreted as a sign of retinal ischemia or microvascular compromise, and has been shown to be enlarged in patients with glaucoma, diabetes and pre-diabetes even without clinical signs of diabetic retinopathy [53,54,55,56]. We hypothesize that these vascular insults can both lead to and/or be caused by metabolic dysfunction of neurons, including RGCs. Reduced blood flow to the central retina can lead to hypoxia, oxidative stress, and impaired metabolic function in RGCs [57,58]. Alternatively, a dysfunctional RGC with a low metabolic rate would require decreased blood supply and therefore decreased capillary density [56]. Regardless of whether the inciting factor is vascular or metabolic, these changes are both accompanied by RGC dysfunction and thus should reduce the amplitude and/or latency (phase) of PERG signals. Therefore, FAZ enlargement may serve as a biomarker of early RGC dysfunction, assessed electro-physiologically via PERG.

Further analysis revealed significant positive correlations between PERG parameters (Mag, MagD, and MagD/Mag ratio) and superior and average RNFL thickness (Table 4). However, FAZ area did not correlate significantly with RNFL thickness in GS. Previous studies in POAG patients reported FAZ enlargement to be associated with RNFL thinning [31,33]. This discrepancy may be explained by the temporal sequence of glaucomatous neurodegeneration. Animal studies have demonstrated that glaucomatous damage begins in the dendrites (IPL) and soma (GCL) before affecting axons (RNFL) [59,60,61,62]. This suggests that RNFL thinning is a later event and may explain why early FAZ enlargement does not correlate with RNFL changes as closely as GCL + IPL changes in GS. Supporting this, our SD-OCT Macular Cube analysis revealed that FAZ area was significantly negatively correlated with GCL + IPL thickness in nearly all macular sectors except the inferior temporal (Table 5).

The cellular response to RGC injury depends on the cumulative damage and the cell’s capacity to adapt and repair, with RGC dysfunction expected to precede programmed cell death [63]. Microvascular insult to the retina, reflected by an enlarged FAZ area, can therefore lead to either RGC dysfunction or cell death [62,63]. We believe that these cellular responses are demonstrated by alterations in Mag and MagD on PERG. Reduction in blood supply to the retina may trigger metabolic stress and eventually initiate apoptotic pathways in RGCs [64,65,66]. The resulting reduction in RGC volume would lead to a reduction in the cumulative electrical activity of RGC signaling and should therefore be reflected by a reduction in Mag [67]. Surviving RGCs, that did not undergo apoptosis, may instead undergo morphological change in response to vascular insult, such as dendritic pruning, soma shrinkage, and axonal thinning [59,68,69]. These structural changes would theoretically increase phase latency, as captured by a reduced MagD [54]. Therefore, FAZ enlargement, as a structural vascular biomarker, likely reflects both cell loss and alterations in RGC morphology and function. The resulting changes in Mag and MagD demonstrates the link between microvascular pathology and electrophysiological impairment in GS.

Additionally, PERG parameters (MAG, MagD, and MagD/Mag ratio) were positively correlated with GCL + IPL thickness, particularly in the superior and superior temporal sectors (Table 5). Macular thickness analysis (Table 6) also demonstrated significant negative correlations between FAZ area and nearly all inner macular sectors, as well as key volumetric indices, including central subfield thickness and total macular volume. Meanwhile, positive correlations were observed between PERG parameters and several inner macular regions.

To further explore the relationship between vascular and structural-functional changes, a mediation analysis was conducted to assess whether FAZ area mediates the effect of MagD on superior GCL + IPL thickness (Figure 4). The analysis revealed that MagD had both a significant direct effect and a significant indirect effect on GCL + IPL thickness through FAZ area. Specifically, higher MagD was associated with smaller FAZ area, which in turn was associated with greater GCL + IPL thickness. These findings suggest that FAZ area may serve as a structural vascular intermediary linking electrophysiological function to retinal morphology. This supports the hypothesis that MagD reflects not only direct neural integrity but also microvascular effects on RGC function and structure. Consistent with this, our team has previously shown that MagD is significantly associated with retinal vessel diameter, a known vascular risk factor for glaucoma [44]. The dual pathway observed in this model reinforces the potential of FAZ area as a biomarker of early glaucomatous changes and highlights the utility of PERG-derived metrics in capturing both functional and vascular contributions to retinal health in GS. Since our analysis demonstrated that FAZ area mediates the relationship between MagD and superior GCL + IPL thickness, we hypothesize that compromised capillary structural integrity in the macula is associated with RGC dysfunction and morphological changes, such as dendritic pruning, soma shrinkage, and axonal thinning, ultimately leading to thinning of the GCL + IPL.

There are several limitations to this study that warrant consideration. The sample size of GS, while sufficient to demonstrate a statistically significant effect, was relatively small and did not include a control group. This may limit the generalizability of our findings to a larger population. Furthermore, the cross-sectional design of our study precludes the establishment of a causal relationship between FAZ area and RGC dysfunction. Longitudinal studies, with serial measurements of FAZ area and RGC function, are crucial to determine the temporal sequence of these changes and to explain their potential predictive value. Additionally, while we attempted to control for potential confounding variables, such as age, sex, CCT, IOP, and spherical equivalent, the possibility of remaining confounding factors cannot be excluded, such as timing of OCTA testing throughout the day to account for any diurnal variation in FAZ area measurements [70]. Other risk factors for FAZ area measurement variation were described in the literature such as smoking hypertension, gender, and alcohol intake [71,72,73].

Despite these limitations, our study provides preliminary, yet compelling, evidence that FAZ enlargement may represent a novel biomarker for early RGC dysfunction in GS. If these findings are corroborated by future longitudinal studies with larger sample sizes, they could have significant clinical implications.

## 5. Conclusions

This study demonstrates that enlargement of the foveal avascular zone (FAZ) is significantly associated with early retinal ganglion cell (RGC) dysfunction in glaucoma suspects (GS), as measured by pattern electroretinogram (PERG). The observed correlations between FAZ area with both functional and structural retinal parameters support the hypothesis that vascular compromise plays a role in early glaucomatous pathophysiology.

## Figures and Tables

**Figure 1 diagnostics-15-02103-f001:**
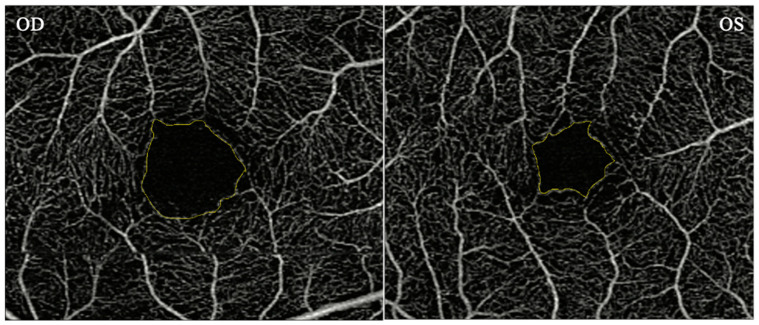
Macular OCT-angiogram demonstrating an enlarged FAZ area (**left image**). Macular OCT-angiogram demonstrating a FAZ area within normal limits (**right image**).

**Figure 2 diagnostics-15-02103-f002:**
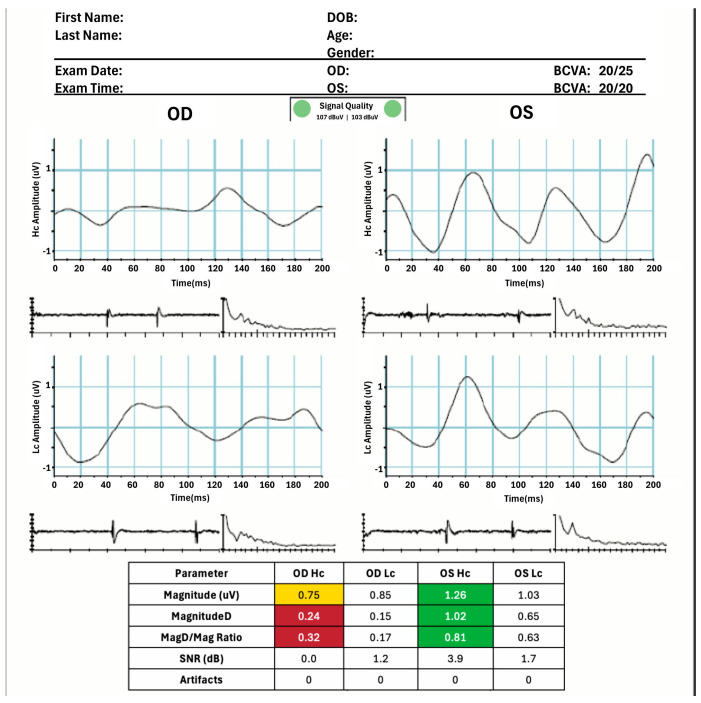
PERG results demonstrating a normal (OS) and an abnormal RGC function (OD). Color coding indicates normal (green), borderline (yellow), and abnormal (red) values.

**Figure 3 diagnostics-15-02103-f003:**
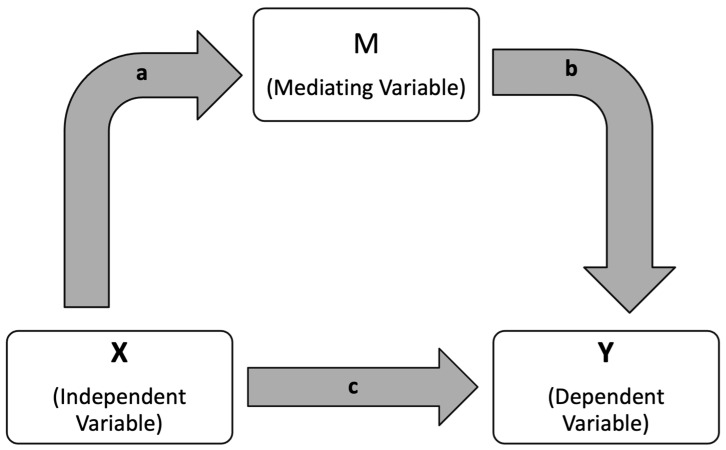
Schematic representation of a mediation models illustrating how an independent variable (X) influences a dependent variables (Y), both directly and indirectly through a mediating variable (M). The path labeled “a” represents the effect of X on M. The paths labeled “b,” represent the effects of M on Y. Both paths “a” and “b” are partial indirect effects. The path labeled “c” represents the direct effect X has on Y.

**Figure 4 diagnostics-15-02103-f004:**
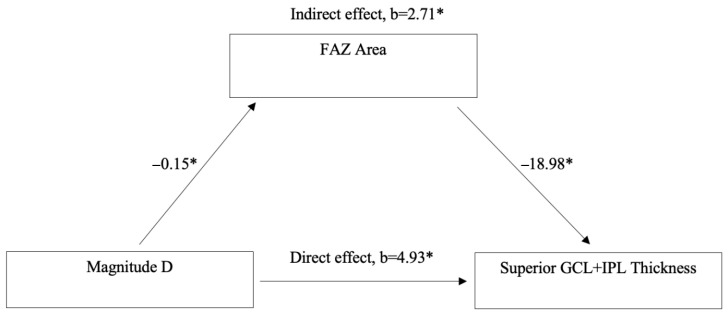
Mediation Analysis: the mediator (M) FAZ area mediating the relationship between MagnitudeD (X) and Superior GCL + IPL Thickness (Y) * *p* < 0.05. Controlled for CCT and Spherical Equivalent. * *p* < 0.05.

**Table 1 diagnostics-15-02103-t001:** Demographic and Clinical Characteristics of Study Subjects (*n* = 20 patients, 31 eyes).

	Mean ± SD
Age (years)	59.89 ± 14.20
Sex	14 Females (70%)
CCT (μm)	551.67 ± 29.22
IOP (mmHg)	18.29 ± 4.46
Spherical Equivalent (D)	−0.531 ± 2.34
Humphrey Field Analyzer	
24-2 MD (dB)	0.21 ± 0.94
24-2 PSD (dB)	1.52 ± 0.58
24-2 VFI (%)	99.25 ± 0.93
ss-PERG	
Magnitude (µV)	1.56 ± 0.43
MagnitudeD (µV)	1.34 ± 0.58
MagnitudeD/Magnitude ratio	0.83 ± 0.11
FAZ Measurements	
FAZ Area (mm^2^)	0.32 ± 0.11
RNFLT (μm)	
Superior RNFLT	106.59 ± 17.7
Temporal RNFLT	67.22 ± 15.01
Inferior RNFLT	118.03 ± 14.30
Nasal RNFLT	70.31 ± 8.64
Average RNFLT	90.50 ± 10.10

SD: Standard Deviation, CCT: Corneal Central Thickness, IOP: Intraocular Pressure, MD: Mean Deviation, PSD: Pattern Standard Deviation, VFI: Visual Field Index, ss-PERG: steady state Pattern Electroretinogram, FAZ: Foveal Avascular Zone, RNFLT: Retinal Nerve Fiber Layer Thickness.

**Table 2 diagnostics-15-02103-t002:** Macular and Ganglion Cell Layer Thickness Characteristics By Sectors (*n* = 20 patients, 31 eyes).

	Mean ± SD
Macular Thickness (μm)	
Superior Outer	277.66 ± 15.12
Superior Inner	320.19 ± 14.21
Nasal Outer	295.66 ± 18.70
Nasal Inner	322.88 ± 14.72
Inferior Outer	268.91 ± 27.36
Inferior Inner	319.38 ± 29.93
Temporal Outer	262.19 ± 18.91
Temporal Inner	311.66 ± 25.98
Thickness Central Subfield	253.91 ± 21.37
Volume Cube (mm^3^)	10.08 ± 0.62
Thickness Average Cube	280.66 ± 16.79
GCL + IPL Thickness (μm)	
Superior	78.00 ± 7.26
Superior Nasal	81.31 ± 10.09
Superior Temporal	77.84 ± 7.43
Inferior	79.66 ± 17.84
Inferior Nasal	81.22 ± 14.91
Inferior Temporal	82.09 ± 14.17
Average GCL + IPL	80.63 ± 10.66
Minimum GCL + IPL	77.06 ± 6.78

SD: Standard Deviation, GCL + IPL: Ganglion Cell Layer + Inner Plexiform Layer.

**Table 3 diagnostics-15-02103-t003:** Partial Correlation of FAZ Area (mm^2^) and PERG Parameters After Controlling for Age, Sex, IOP, CCT, and Spherical Equivalent.

	FAZ Area
Magnitude (μV)	−0.503 *
MagnitudeD (μV)	−0.507 *
MagD/Mag ratio	−0.293

FAZ: Foveal Avascular Zone, PERG: Pattern Electroretinogram, MagD/Mag: MagnitudeD/Magnitude ratio, IOP: Intraocular Pressure, CCT: Central Corneal Thickness. * *p* < 0.05.

**Table 4 diagnostics-15-02103-t004:** Partial Correlation of RNFLT (μm), FAZ Area (mm^2^) & PERG parameters After Controlling for Age, Sex, IOP, CCT and Spherical Equivalent.

	Superior	Temporal	Inferior	Nasal	Average
Mag (μV)	0.759 **	0.187	0.388	0.131	0.537 *
MagD (μV)	0.807 **	0.255	0.391	0.251	0.600 *
MagD/Mag ratio	0.628 *	0.253	0.292	0.417	0.530 *
FAZ Area (mm^2^)	−0.273	−0.034	−0.136	0.008	−0.165

RNFLT: Retinal Nerve Fiber Layer Thickness, FAZ: Foveal Avascular Zone, PERG: Pattern Electroretinogram, Mag: Magnitude, MagD: MagnitudeD, MagD/Mag ratio: MagnitudeD/Magnitude ratio, IOP: Intraocular Pressure, CCT: Central Corneal Thickness. * *p* < 0.05. ** *p* < 0.001.

**Table 5 diagnostics-15-02103-t005:** Partial Correlation Analysis of GCL + IPL (μm), FAZ Area (mm^2^) and PERG Parameters after Controlling for Age, Sex, IOP, CCT, Spherical Equivalent.

	GCL + IPL Thickness Sectors (μm)						Average GCL + IPL (μm)	Minimum GCL + IPL (μm)
	Superior	Superior Nasal	Superior Temporal	Inferior	Inferior Nasal	Inferior Temporal		
Mag (μV)	0.586 *	0.434	0.556 *	0.246	0.496 *	0.249	0.417	0.452
MagD (μV)	0.618 *	0.389	0.621 *	0.287	0.452	0.335	0.490 *	0.540 *
MagD/Mag ratio	0.527 *	0.225	0.599 *	0.412	0.289	0.448	0.595 *	0.704 *
FAZ Area (mm^2^)	−0.688 *	−0.662 *	−0.507 *	−0.533 *	−0.686 *	−0.311	−0.286	−0.195

GCL + IPL: Ganglion Cell + Inner Plexiform Layer, FAZ: Foveal Avascular Zone, PERG: Pattern Electroretinogram, Mag: Magnitude, MagD: MagnitudeD, MagD/Mag ratio: MagnitudeD/Magnitude ratio, IOP: Intraocular Pressure, CCT: Central Corneal Thickness. * *p* < 0.05.

**Table 6 diagnostics-15-02103-t006:** Partial Correlation of Macular Thickness (μm), FAZ Area (mm^2^) and PERG Parameters After Controlling for Age, Sex, IOP, CCT and Spherical Equivalent.

	Superior Outer	Superior Inner	Nasal Outer	Nasal Inner	Inferior Outer	Inferior Inner	Temporal Outer	Temporal Inner	Thickness Central Subfield	Volume Cube (mm^3^)	Thickness Average Cube
Mag (μV)	0.341	0.545 *	0.483 *	0.517 *	0.161	0.547 *	0.370	0.594 *	0.267	0.364	0.353
MagD (μV)	0.357	0.512 *	0.455	0.451	0.212	0.481 *	0.420	0.625 *	0.265	0.387	0.391
MagD/Mag ratio	0.250	0.288	0.234	0.173	0.186	0.196	0.376	0.509 *	0.085	0.266	0.347
FAZ Area (mm^2^)	−0.555 *	−0.580 *	−0.708 *	−0.655 *	−0.434	−0.562 *	−0.371	−0.512 *	−0.761 **	−0.631 *	−0.471 *

FAZ: Foveal Avascular Zone, PERG: Pattern Electroretinogram, Mag: Magnitude, MagD: MagnitudeD, MagD/Mag ratio: MagnitudeD/Magnitude ratio, IOP: Intraocular Pressure, CCT: Central Corneal Thickness. * *p* < 0.05. ** *p* < 0.001.

## Data Availability

The data presented in this study are available on request from the corresponding author due to patient privacy.

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
