# Peer review of "Foveal Avascular Zone Enlargement as a Risk Factor for Early Retinal Ganglion Cell Dysfunction in Glaucoma Suspects"

_diagnostics, 2025, doi:10.3390/diagnostics15162103_

Round 1

Reviewer 1 Report

Comments and Suggestions for Authors

I would like to congratulate the authors on their really interesting paper. This is an innovative work that links microvascular foveal integrity with early retinal ganglion cell dysfunction in glaucoma suspect patients. A few issues that could further improve the quality of the paper:

a. The authors can add that all OCTA measurements were captured during the same time of the day to account for any diurnal variation that might affect the foveal avascular measurement indices [1].

b. The authors can add that FAZ area has been found to be enlarged even in patients with pre diabetes or patients with diabetes without any evident signs of diabetic retinopathy in line 385 [2,3].

1.Lal B, Alonso-Caneiro D, Read SA, Tran B, Van Bui C, Tang D, Fiedler JT, Ho S, Carkeet A. Changes in Retinal Optical Coherence Tomography Angiography Indexes Over 24 Hours. Invest Ophthalmol Vis Sci. 2022 Mar 2;63(3):25. doi: 10.1167/iovs.63.3.25. PMID: 35348589; PMCID: PMC8976927.

2.Kazantzis D, Holmes C, Wijesingha N, Sivaprasad S. Changes in foveal avascular zone parameters in individuals with prediabetes compared to normoglycemic controls: a systematic review and meta-analysis. Eye (Lond). 2024 Jul;38(10):1855-1860. doi: 10.1038/s41433-024-03058-5. Epub 2024 Apr 8. PMID: 38589460; PMCID: PMC11226666.

3.Altinisik M, Kahraman NS, Kurt E, Mayali H, Kayikcioglu O. Quantitative analysis of early retinal vascular changes in type 2 diabetic patients without clinical retinopathy by optical coherence tomography angiography. Int Ophthalmol. 2022 Feb;42(2):367-375. doi: 10.1007/s10792-022-02230-8. Epub 2022 Jan 31. PMID: 35099665.

Author Response

Comment 1: The authors can add that all OCTA measurements were captured during the same time of the day to account for any diurnal variation that might affect the foveal avascular measurement indices.

Response 1: We thank the reviewer for highlighting the potential impact of diurnal variation on FAZ measurements. We have now acknowledged this limitation in the revised discussion section on line 460.

Comment 2: The authors can add that FAZ area has been found to be enlarged even in patients with pre-diabetes or patients with diabetes without any evident signs of diabetic retinopathy in line 385.

Response 2: We appreciate this suggestion. As recommended, we have revised the Discussion section to include that FAZ enlargement has been observed in patients with diabetes and pre-diabetes even in the absence of clinical signs of diabetic retinopathy (line 385-386).

Reviewer 2 Report

Comments and Suggestions for Authors

Summary:

This cross-sectional study examines the correlation between foveal avascular zone (FAZ) enlargement measured by optical coherence tomography angiography (OCTA), retinal ganglion cell (RGC) dysfunction measured by pattern electroretinogram (PERG), and retinal nerve fiber layer (RNFL) and ganglion cell layer-inner plexiform layer (GCL-INL) thinning measured by optical coherence tomography (OCT) in individuals with glaucoma suspicion. The study included 31 eyes from 20 glaucoma suspects (GS) subjects, diagnosed based on increased cup-to-disc ratio, rim thinning, notching, or excavation. After a thorough ophthalmologic evaluation to confirm the GS status, the subjects underwent the examinations listed above. Partial correlation analysis was done with parameters to evaluate: 1) whether GS with retinal ganglion cell (RGC) dysfunction were associated with FAZ enlargement, and 2) whether FAZ enlargements were correlated with RGC dysfunction and structural retinal thinning (i.e. retinal nerve fiber layer (RNFL) and ganglion cell layer-inner plexiform layer (GCL-IPL) thinning). The results demonstrated that enlargement of the FAZ area was significantly correlated with RGC dysfunction. Although FAZ enlargement was not associated with RNFL thinning, it was significantly correlated with GCL-IPL thinning across all regions except the inferior temporal sector. Similarly, FAZ enlargement was associated with reduced thickness in all macular regions, with the exception of the inferior and temporal outer sectors. RGC dysfunction also showed significant negative correlations with superior and average RNFL thickness, superior and superior temporal GCL-IPL thickness, and all macular regions except the inferior and temporal outer sectors. Furthermore, mediation analysis revealed that FAZ enlargement partially mediates the relationship between RGC dysfunction and superior GCL-IPL thinning. Collectively, these findings suggest that FAZ enlargement may serve as a potential biomarker for earlier detection of RGC dysfunction in glaucoma suspect patients, allowing earlier management before permanent vision loss.

Strengths:

  • The author studies FAZ as a possible biomarker for glaucoma detection. OCTA is non-invasive and already implemented in ophthalmic practices. Therefore, raising FAZ analysis as a potential early diagnostic tool in at-risk or glaucoma suspect populations is quite relevant, especially if glaucomatous changes can be observed before functional damage (visual field loss was an exclusion criterion).

Suggestions:

  • Given the already limited dataset, the study may consider an asymmetric CD ratio (difference >0.2) as another inclusion criterion.
  • The partial correlation analysis with significance evaluation findings provides strong evidence that there is a correlation between FAZ area enlargement, RGC dysfunction, and structural thinning in glaucoma suspects. However, in the context of a small sample size and with less stringent inclusion criteria, a control group becomes important to evaluate whether FAZ enlargement is truly caused by pre-glaucomatous pathologies.

Author Response

Comment 1: Given the already limited dataset, the study may consider an asymmetric CD ratio (difference >0.2) as another inclusion criterion.

Response 1: As the reviewer recommended, we adapted a 0.2 difference in asymmetric C/D ratio between both eyes of a subject, and this change did not lead to any significant change in the results.

Comment 2: The partial correlation analysis with significance evaluation findings provides strong evidence that there is a correlation between FAZ area enlargement, RGC dysfunction, and structural thinning in glaucoma suspects. However, in the context of a small sample size and with less stringent inclusion criteria, a control group becomes important to evaluate whether FAZ enlargement is truly caused by pre-glaucomatous pathologies.

Response 2: We agree with the reviewer’s assessment. As noted in the revised Discussion section, the relatively small sample size and absence of a control group are limitations of this study. We have updated the discussion section and acknowledged this in line 452.